# PromptNER: Prompting For FewShot Named Entity Recognition

## Abstract

In a surprising turn, Large Language Models (LLMs), together with a growing arsenal of prompt-based heuristics, provide powerful few-shot solutions to myriad classic NLP problems. However, despite promising early results, current LLM-based few-shot methods remain far from the state of the art in Named Entity Recognition (NER), where prevailing methods include learning representations via end-to-end structural understanding and fine-tuning on standard labeled corpora. In this paper, we introduce PromptNER, an algorithm for few-shot and cross-domain NER. To adapt to a new NER task, PromptNER requires *a set of entity definitions*, and a set of few-shot examples, along with explanatory text justifying the applicability of each entity tag. Given a sentence, PromptNER prompts an LLM to produce a list of potential entities along with corresponding explanations justifying their compatibility with the provided entity type definitions. PromptNER achieves state-of-the-art performance on few-shot NER, achieving improvements in F1 score (absolute) of 4% on the ConLL dataset, 9% on GENIA, 4% on FewNERD, 5% on FaBNER and 24% on TweetNER. PromptNER also achieves state-of-the-art performance on Cross Domain NER beating even methods not restricted to the few-shot setting on $3/5$ CrossNER target domains, with an average F1 gain of 3%, despite using less than 2% of the available data.

## 1 Introduction

Named Entity Recognition (Chinchor, 1995) is often a vital component in text processing pipelines for information extraction and semantic understanding (Sharma et al., 2022; Ali et al., 2022). Current methods perform well when training data is plentiful (Wang et al., 2022; Yu et al., 2020; Li et al., 2022; Wang et al., 2020). However, their applicability to many real-world problems is hindered by their reliance on fixed entity definitions and large amounts of in-domain training data for the specific NER formulation and population of interest. Unfortunately, commitments about what constitute the relevant entities vary wildly across use cases, a fact reflected in the diversity of academic datasets (contrast, e.g., medical NER datasets with CoNLL or OntoNotes). Ultimately, these differing commitments stem from differences in the envisioned use cases. Should we categorize the phrase 'Theory of General Relativity' as an entity? A media company tasked with extracting information from political articles might not designate physical laws as a relevant class of entities but a scientific journal might. Given such diversity of use cases, we cannot hope for a universal NER system. However, we might hope to produce NER systems capable of adapting to new settings flexibly, requiring only modest human effort.

With the emergence of LLMs, the NLP community has developed a repertoire of in-context learning strategies that have rapidly advanced the state of few-shot learning for myriad tasks (Brown et al., 2020; Wei et al., 2022; Liu et al., 2023). However, such prompting-based approaches have yet to show comparable impact in NER, where current methods typically cast few-shot learning as a domain transfer problem, training on large amounts of source data and fine-tuning on exemplars from the target domain (Huang et al., 2022b; Yang et al., 2022). Moreover, a significant gap remains between the best few-shot NER methods and the performance of end-to-end trained models (Wang et al., 2022; Xu et al., 2022). These few-shot methods also struggle when the source and target domains differ with respect to what constitutes an entity (Yang et al., 2022; Das et al., 2022). A separate class of adaptation methods have shown promise when the source and target vary considerably,

---

**Input**

Defn: An entity is a person(person), university(university), scientist(scientist), organisation(organisation), country(country), location(location), scientific discipline(discipline), enzyme(enzyme), protein(protein), chemical compound(chemicalcompound), chemical element(chemicalelement), event(event), astronomical object(astronomicalobject), academic journal(academicjournal), award(award), or theory(theory). If an entity does not fit the types above it is (misc)

Example 1: He attended the U.S. Air Force Institute of Technology for a year , earning a bachelor 's degree in aeromechanics , and received his test pilot training at Edwards Air Force Base in California before his assignment as a test pilot at Wright-Patterson Air Force Base in Ohio .

Answer:
1. U.S. Air Force Institute of Technology | True | as he attended this institute is likely a university (university)
2. bachelor 's degree | False | as it is not a university, award or any other entity type
3. aeromechanics | True | as it is a scientific discipline (discipline)
4. Edwards Air Force Base | True | as an Air Force Base is an organised unit (organisation)
5. California | True | as in this case California refers to the state of California itself (location)
6. Wright-Patterson Air Force Base | True | as an Air Force Base is an organisation (organisation)
7. Ohio | True | as it is a state (location)

Figure 1: Example of prompt to pre-trained language model. Definition is in blue, question and task in red, example answer and chain of thought format in green

but they tend to require hundreds of data points to be effective (Hu et al., 2022a; Chen et al., 2023; Hu et al., 2022b; Chen et al., 2022).

In this paper, we introduce PromptNER, a prompt-based NER method that achieves state-of-the-art results on few-shot NER and CrossDomain NER. Our method consists of 3 key components—a backbone LLM, a modular definition (a document defining the set of entity types), and a precise format for outputting the extracted entities, which is communicated to the model via the formatting of the few-shot from the target domain. To adapt to a new domain, our method requires (i) the entity definitions; (ii) the few-shot examples; and (iii) a set of explanations justifying the applicability of each entity tag. The process is fully automated but for two steps—the crafting of the definition and the explanations for the few shot examples. This makes the method flexible and easy to apply across domains. In all experiments, we simply copy paste the entity definitions from the documentation accompanying each dataset. The explanations were hand-written by the author without any reference to performance on a validation set.

**Key results**: PromptNER achieves 83.48% F1 score on the CoNLL dataset (Sang & De Meulder, 2003) in a few-shot setting, improving over the best previous few-shot methods by 4% (absolute). PromptNER outperforms the best-competing methods by 9% on the GENIA (Kim et al., 2003) dataset, 4% on the FewNERD-Intra (Ding et al., 2021) setting, 5% on FaBNER (Kumar & Starly, 2022), and 24% on TweetNER (Ushio et al., 2022) and achieves state-of-the-art numbers on three out of five of the CrossNER (Liu et al., 2021) target domains, despite using only 2% of the available training data. In Sections 4 and 5, we show that this strong performance is not due to data contamination and that PromptNER consistently outperforms few-shot Prompting (Brown et al., 2020) and Chain-of-Thought Prompting baselines (Wei et al., 2022).

## 2 BACKGROUND AND RELATED WORKS

**Named Entity Recognition** has been well studied since the formulation of the problem in the mid-90s (Chinchor, 1995), with early methods including rule-based systems (Farmakiotou et al., 2000; Mikheev et al., 1999) and statistical learning based methods (Borthwick et al., 1998; Borthwick, 1999). With the shift towards deep learning, RNN and transformer-based methods came to domi-

nate on most NER tasks (De Mulder et al., 2015). Most recently, methods leveraging pre-trained Transformers have advanced the state-of-the-art further (Lu et al., 2022; Wang et al., 2022).

Recent methods include DeepStruct (Wang et al., 2022), which modifies the language modelling objective to be more aware of logical structure in the corpus. Other methods introduce specialized architectures for NER (Yu et al., 2020; Li et al., 2022). These methods typically require full dataset access and require significant computational resources for training. Motivated by settings where training on full datasets is not possible or practical, some researchers have turned their attention to few-shot NER (Church et al., 2021; Das et al., 2022; Huang et al., 2022b). Leading few-shot NER methods include approaches which seek to create a pretrained LLM that can then be adapted for NER (Wang et al., 2022; Xu et al., 2022) and approaches which view NER as a metric learning problem and use prompts to guide the learning process (Huang et al., 2022a; Chen et al., 2022). Some methods tackle the setting where the training and testing tasks are from different domains, these include CP-NER (Chen et al., 2023), which uses collaborative prefix tuning to learn domain-specific prefixes that can be swapped flexibly to perform NER; FactMix (Yang et al., 2022), which uses a model-agnostic data augmentation strategy to improve generalization; and LANER (Hu et al., 2022a), which tries to improve transferability of learnt label information. A few recent efforts apply in-context learning (Wei et al., 2021) directly for NER, often interpreting the problem as a templated cloze statement infilling problem (Lee et al., 2021; Cui et al., 2021).

**Prompting and Chain-of-Thought Prompting**: Brown et al. (2020) demonstrated that LLMs can achieve high performance on few-shot tasks simply by prepending the few-shot examples to the input. Chain-of-thought prompting (Wei et al., 2022) extends this by providing examples in the prompt which contain both question-answer pairs and some stated reasoning for the provided answer.

## 3 METHOD

Our method consists of 3 key components (Figure 1):

**Conditional Generation Language Model**: we leverage the power of pretrained LLMs which have been trained on conditional generation tasks. This is a departure from several works in the field that consider the NER problem as a sequence to label (or discriminative) problem (Ali et al., 2022; Sharma et al., 2022), or the prompting-based approaches that consider NER to be a cloze statement prediction problem (Cui et al., 2021; Lee et al., 2021). Using general Seq-to-Seq models opens up modeling possibilities, allowing us to pursue strategies based on chain-of-thought-like reasoning. These possibilities are not readily available in the aligned sequence tagging architectures traditionally used for NER.

**Modular Definitions:** the implicit commitments about what constitute instances of the relevant entity types can vary wildly across different NER settings. The difficulty of capturing these subtleties in just a few examples can hinder typical few-shot approaches to NER. In our approach to few-shot, NER, each problem is defined not only by a small set of exemplars (the few-shot examples) but also by a per-domain definition. Here, the modular definition consists of a natural language description of what does (and does not) constitute an entity. This can be useful in instances where the typical natural language connotation of the word 'entity' may include concepts the specific NER task would want to exclude. Since this document is composed in natural language, it can easily be composed by an end user with no technical knowledge. In all our experiments, the definitions are the list of entity types copy pasted from the documentation of each dataset. We later show that tuning this definition leads to significant gains (Section 5).

**Potential Entity Output Template:** Motivated by Wei et al. (2022), we create a template structure for the output of the LLM which allows it to emulate reasoning and justify why a phrase is predicted as an entity of a specific type. The exact structure is one where each line of the output mentions a distinct candidate entity, a decision on whether or not the candidate should be considered an entity and an explanation for why or why not along with what entity type it belongs to. In the ablation section, we show that the inclusion of examples and a clear definition are the most important parts of this NER system.

| Method | CoNLL |
|---|---|
| COPNERHuang et al. (2022b) | $75.8 \pm 2.7$ |
| EntLMMa et al. (2021) | $51.32 \pm 7.67$ |
| FactMixYang et al. (2022) | 60.8 |
| ProMLChen et al. (2022) | $79.16 \pm 4.49$ |
| UIELu et al. (2022) | 67.09 |
| CONTaiNERDas et al. (2022) | $75.8 \pm 2.7$ |
| PMRXu et al. (2022) | $65.7 \pm 4.5$ |
| PromptNER T5XXL (Us) | $45.66 \pm 12.43$ |
| **PromptNER GPT3.5 (Us)** | $78.62 \pm 4.62$ |
| **PromptNER GPT4 (Us)** | $\mathbf{83.48 \pm 5.66}$ |

Table 1: FewShot Learning ($0 < k < 5$) on CoNLL dataset. Results show micro-F1 averages and associated standard deviation over 3 runs when available.

Seen as a whole, this pipeline provides flexibility with little cost. The only components that need to change to adapt the method to new domains are the definition and examples, both of which have virtually no computational cost and do not require an entire data collection initiative to gather.[1]

## 4 EXPERIMENTS AND RESULTS

**Model Description and Prompt Engineering:** For all the experiments below we present the result of our method when using T5-Flan (11B) (Chung et al., 2022), GPT-3.5 (Brown et al., 2020) (text-davinci-003 model), and GPT4 (OpenAI, 2023). The results for all competing methods are taken from the tables reported in their respective publications and papers. We use the standard Micro-F1 metric in NER (De Mulder et al., 2015; Wang et al., 2022) and report the mean and variance over 3 runs on the test set.

To minimize the effect of prompt engineering, the definitions of the entity types for each dataset is the list of entity types and descriptions of each type taken exactly from the papers of the respective datasets. The few-shot examples are randomly selected support sets from the training split. The non-entity phrases, which are candidate entities in the few-shot demonstrations, are selected at random after removing stop words. The explanations for why an entity is of a particular type is written by the researchers. In the next section, we investigate the robustness of the results to different annotators and explore a fully automatic system where the LLM itself creates the explanations for the few-shot demonstrations.

**Standard Low Resource NER:** In this experiment we used the most common NER dataset: CoNLL (Sang & De Meulder, 2003). For this dataset, standard methods can reach F1 scores in the range of $91\% - 94\%$ (Wang et al., 2022). However, when used in low-resource settings, the available methods are significantly less powerful. We show results for few-shot Learning with $k = 5$ points from CoNLL for all the competing methods and report the averaged micro-F1 scores of the models. PromptNER outperforms all competing methods when evaluated in the low-resource regime—with GPT4 achieving absolute performance gains of around $4\%$ in F1 score (Table 1).

**Cross Domain NER:** In this experiment, we use the CrossNER (Liu et al., 2021) dataset. The training set is a subset of CoNLL, but the evaluation domains are from five domains with different entity types—Politics, Literature, Music, AI, and Natural Sciences. The conception of what is an entity can vary significantly across domains. For example, in the AI split, abstract mathematical algorithms and methods in machine learning like the phrase 'deep learning' are considered entities, while in politics, abstract methods like 'polling' are not entities. On this dataset, few-shot methods are not typically able to perform well with only around 5-10 examples, and hence all other successful methods use a significant portion of the training and dev splits of CrossNER (100-200 examples). Despite using only $1\% - 2\%$ of the data as the other methods, we are able to achieve state-of-the-art performance in 3/5 of the domains with GPT4, outperforming other methods by an absolute F1 of $3\%$ on average (Table 2). All PromptNER systems perform worse on the AI and Science domain.

---

[1]Code available at `https://anonymous.4open.science/r/PromptNER-2BB0/`

| Method | k | Politics | Literature | Music | AI | Sciences |
|---|---|---|---|---|---|---|
| FactMixYang et al. (2022) | 100 | 44.66 | 28.89 | 23.75 | 32.09 | 34.13 |
| LANERHu et al. (2022a) | 100-200 | 74.06 | 71.11 | 78.78 | 65.79 | 71.83 |
| CPNERChen et al. (2023) | 100-200 | 76.35 | 72.17 | 80.28 | **66.39** | **76.83** |
| EnTDAHu et al. (2022b) | 100 | 72.98 | 68.04 | 76.55 | 62.31 | 72.55 |
| PromptNER T5XXL (Us) | 2 | 39.43 | 36.55 | 41.93 | 30.67 | 46.32 |
| **PromptNER GPT3.5 (Us)** | 2 | 71.74 | 64.15 | 77.78 | 59.35 | 64.83 |
| **PromptNER GPT4 (Us)** | 2 | **78.61** | **74.44** | **84.26** | 64.83 | 72.59 |

Table 2: Cross Domain results on CrossNER dataset with CoNLL as source domain. $k$ is the number of target domain datapoints used by each method. Results show micro-F1 scores. Despite using only 1%–2% of the data our method achieves state-of-the-art performance on three of the five datasets

| Method | GENIA |
|---|---|
| CONTaiNERDas et al. (2022) | $44.77 \pm 1.06$ |
| BCLMing et al. (2022) | $46.06 \pm 1.02$ |
| SpanProtoShen et al. (2021) | $41.84 \pm 2.66$ |
| PACL | $49.58 \pm 1.82$ |
| PromptNER T5XXL (Us) | $25.13 \pm 3.22$ |
| **PromptNER GPT3.5 (Us)** | $\mathbf{52.80 \pm 5.15}$ |
| **PromptNER GPT4 (Us)** | $\mathbf{58.44 \pm 6.82}$ |

Table 3: Few-shot Learning ($0 < k < 5$) on GENIA dataset. Results show micro-F1 averages and associated standard deviation over 3 runs when available.

**Biomedical Domain NER:** We next use the GENIA dataset (Kim et al., 2003), a biomedical dataset constructed using PubMed (White, 2020). This presents a different domain with a significant shift in vocabulary in the corpus when compared to the previous datasets. This dataset also has more entity types (32 vs 17) and is more technical than the CrossNER Natural Sciences domain. We use the complete 32-way 5-shot setting. Once again, PromptNER outperforms all competing methods, with GPT3.5 outperforming the best competitor by 3% absolute F1 and GPT4 doing so by 9% (Table 3).

**Contradictory Domain NER:** In this experiment, we use the intra-split of the FewNERD dataset (Ding et al., 2021), using the test split and compiling results for the 10-way problem setting. This dataset is contradictory, in that the sets marked train, dev, and test all have non-overlapping entity types which are labeled as entities. For example, while the train split considers people as entities but not events or buildings, the dev set considers only buildings or events to be entities. This is a difficult benchmark as the labels of the training and dev sets are actively misleading when the goal is to perform well on the test set. Our method outperforms all competing methods in this setting, doing so by an average (absolute) percentage increase of over 4% F1 (Table 4). This shows the flexibility of our method in even the most pathological of cases: where the requirement for entities to be extracted is actively in conflict with the common understanding of the word 'entity', and the existing data sources. Changing the definition to explicitly rule out objects which would normally be considered an entity is particularly useful here, as shown in later ablations (Section 5).

| Method | k | FewNERD |
|---|---|---|
| ProMLChen et al. (2022) | 5 | $68.1 \pm 0.35$ |
| CONTaiNERDas et al. (2022) | 5 | 47.51 |
| Meta LearningMa et al. (2022) | 5 | $56.8 \pm 0.14$ |
| PromptNER T5XXL (Us) | 2 | $55.7 \pm 1.09$ |
| **PromptNER GPT3.5 (Us)** | 2 | $62.33 \pm 6.30$ |
| **PromptNER GPT4 (Us)** | 2 | $\mathbf{72.63 \pm 5.48}$ |

Table 4: Few-shot results on the FewNERD dataset on the INTRA 10-way task, $k$ is the number of datapoints used by each method. Results show micro-F1 scores.

| Methods | TweetNER | FaBNER |
|---|---|---|
| CONTaiNER | 12.83 ± 6.3 | 9.42 ± 5.1 |
| ProML | 18.82 ± 4.2 | 19.61 ± 3.2 |
| PromptNER GPT3.5 | 30.5 ± 5.6 | 17.84 ± 4.8 |
| PromptNER GPT4 | **43.5 ± 5.3** | **24.35 ± 4.1** |

Table 5: Performance of PromptNER on TweetNER and FaBNER. GPT3.5 outperforms both methods significantly on TweetNER, and is competitive with all other methods on FaBNER.

| Model | Size | ConLL | Genia | Politics | Literature | Music | AI | Science | FewNERD |
|---|---|---|---|---|---|---|---|---|---|
| GPT4 | **1.8T\*** | **83.48** | **58.44** | **78.61** | **74.44** | **84.26** | **64.83** | **72.59** | **72.63** |
| GPT3 | 175B | 78.62 | 52.8 | 71.74 | 64.15 | 77.78 | 59.35 | 64.83 | 62.33 |
| T5XXL | 11B | 45.66 | 19.34 | 39.43 | 36.55 | 41.93 | 30.67 | 46.32 | 23.2 |
| T5XL | 3B | 24.12 | 10.5 | 18.45 | 18.62 | 25.79 | 10.58 | 26.39 | 8.35 |

Table 6: Model performance over various model sizes, there are clear benefits to scaling the backbone Large Language Model. * GPT4 model size is unconfirmed.

**Recent NER Datasets:** Since we are using Large Language Models, which have been trained on data from undisclosed sources (Brown et al., 2020; OpenAI, 2023), we consider the possibility that parts of the evaluation set from the above experiments have been seen by the model during the language modeling period. Recent results have shown instances where the performance of Few-Shot Prompting on contaminated data (which has been seen during training) is not significantly different from the performance on clean, unseen data (Chowdhery et al., 2022). However, to more conclusively show that the improved performance of PromptNER is not a function of data contamination, we show results for two NER datasets released after the GPT3.5 cutoff date. TweetNER (Ushio et al., 2022) (test 2021 split) is a dataset of tweets with 7 entity types, and FaBNER (Kumar & Starly, 2022) is a dataset focused on manufacturing and fabrication science with 12 different types. The datasets also have diverse domains, with TweetNER having Twitter handles, URL names and other nonstandard entities while FaBNER has no entity type overlaps with any other dataset studied in this paper. We compare our method to the most successful few-shot methods from Table 1, CONTaiNER (Das et al., 2022) and ProML (Chen et al., 2022). PromptNER GPT3.5 outperforms all methods by a significant margin on TweetNER, achieving an F1 score improvement of 12%(Table 5). On FaBNER, however, PromptNER GPT3.5 underperforms when compared to ProML. The GPT4 model achieves state-of-the-art performance on both datasets. Since both datasets were released after the information cutoff of GPT3.5, we can be sure that the superior performance of GPT3.5 is not because the labels of the task were in some way used during the language modeling phase.

## 5 ABLATIONS AND INVESTIGATIONS

In this section, we set out to investigate the effects of the different aspects of this pipeline and answer some questions on what really matters to the success of the method: In all the tables for this section, we refer to the CrossNER Domains by their domain name alone and we refer to the FewNERD Test Intra 10 way setting as FewNERD alone.

**Pretrained Language Model:** We hold all other parts of the pipeline constant (the definitions, examples, and chain of thought structure of the output) and vary the baseline model that we use to compute our predictions, to understand whether there is any trend over which models perform better. As expected, there are significant gains to scaling the size of the Large Language Model (Table 6). A qualitative analysis of the results suggests that T5XL is barely able to perform the instruction provided in the prompt, sometimes breaking the output format structure, often predicting a single entity multiple times, etc. T5XXL is much better, it is able to follow the output format consistently. However, it is unable to use the definition of entities properly, often labeling dates, numbers, and months as entities despite being explicitly provided a definition that excludes these types of words. This gives us reason to believe that this method is likely to improve as LLMs get better at following instructions more exactly.

| Def | FS | CoT | Cand | ConLL | Genia | Pol | Lit | Mus | AI | Sci | FewNERD | Avg Rank |
|-----|----|----|----|-------|-------|-----|-----|-----|----|-----|---------|----------|
| ✓ | ✓ | ✓ | ✓ | **78.6** | **52.8** | **71.7** | **64.1** | **77.7** | **59.3** | **64.8** | **62.3** | 1 |
| ✓ | ✓ | ✓ | ✗ | 71.6 | 38.5 | 61.3 | 46.3 | 60.2 | 34.2 | 46.8 | 57.3 | 3.5 |
| ✓ | ✓ | ✗ | ✓ | 75.1 | 49.2 | 70.4 | 54.9 | 70.6 | 53.6 | 60.5 | 42.4 | 2.1 |
| ✓ | ✗ | ✓ | ✓ | 68.1 | 23.2 | 20.3 | 21.3 | 24.5 | 40.7 | 40.6 | 34.6 | 5.6 |
| ✗ | ✓ | ✓ | ✓ | 63.3 | 46.2 | 57.7 | 49.6 | 50 | 29 | 50.8 | 34.8 | 4 |
| ✗ | ✓ | ✓ | ✗ | 54.8 | 37.2 | 49.8 | 37.3 | 54.7 | 27.8 | 21.7 | 18.8 | 5.6 |
| ✗ | ✓ | ✗ | ✗ | 49.7 | 39.3 | 42.5 | 40.3 | 48.6 | 24.5 | 35.9 | 16.1 | 6.1 |

Table 7: Ablation over components of PromptNER on GPT3.5. Def: Definitions, FS: Few Shot Examples, CoT: Explanations required, Cand: Candidate entities in predicted list. Every component improves performance of the method in general, with the setting of all components vastly outperforming the traditional Few Shot Prompting and Chain-of-Thought Prompting methods

| Def | FS | CoT | Cand | ConLL | Genia | Pol | Lit | Mus | AI | Sci | FewNERD | Avg Rank |
|-----|----|----|----|-------|-------|-----|-----|-----|----|-----|---------|----------|
| ✓ | ✓ | ✓ | ✓ | 83.4 | **58.4** | **78.6** | **74.4** | **84.2** | 64.8 | **72.5** | **72.6** | 1.2 |
| ✓ | ✓ | ✓ | ✗ | 78.5 | 51.8 | 70.3 | 69.6 | 73.8 | **66.1** | 67.6 | 59.5 | 2.5 |
| ✓ | ✓ | ✗ | ✓ | 67.4 | 49.2 | 72.4 | 63.5 | 80.5 | 60.5 | 59.7 | 62.8 | 3 |
| ✓ | ✗ | ✓ | ✓ | **84.3** | 30.2 | 62.8 | 53.2 | 63.5 | 42.7 | 41.4 | 30.2 | 4.7 |
| ✗ | ✓ | ✓ | ✓ | 70.7 | 53.3 | 65.6 | 55 | 53 | 45.1 | 53.7 | 43.2 | 4.1 |
| ✗ | ✓ | ✓ | ✗ | 63 | 38.5 | 60.2 | 54.6 | 58.9 | 35.9 | 46.3 | 27.1 | 5.6 |
| ✗ | ✓ | ✗ | ✗ | 66.7 | 27.4 | 58.3 | 46.4 | 54.7 | 27.7 | 36.2 | 21.9 | 6.2 |

Table 8: Ablation over components of CoTNER on GPT4. Def: Definitions, FS: Few Shot Examples, CoT: Explanations required, Cand: Candidate entities in predicted list. Every component improves performance of the method in general, with the setting of all components vastly outperforming traditional Few Shot Prompting and Chain-of-Thought Prompting. Candidate inclusion is the single most important component

**Components of PromptNER:** We can consider PromptNER as having 4 different components that can sequentially be turned off - the provision of a definition (Defn), the provision of few-shot examples (FS), the requirement to explain the reasoning on why a candidate is or is not an entity (CoT) and finally whether or not the list should contain only entities, or also contain candidate entities that may be declared as not meeting the definition of an entity (Cand). We study how the performance changes for GPT3.5 as we remove only one component of the system and specifically check configurations of Chain-of-thought Prompting and FewShot Prompting (Table 7). For the sake of brevity, we only show the mean of the Micro-F1, however, a complete table with standard deviations can be found in the appendix (Table 13). The results consistently show that every part of the pipeline is useful, with the definition being more important in tasks where the common conception of an entity is quite different from that of the domain (FewNERD, AI) as opposed to the more standard setting (ConLL, Politics). Notably, the row that corresponds to classic Few Shot Learning has an average rank of 6.1, with classic Chain of Thought Prompting having an average rank of 5.6, both of these are greatly outperformed by the setting with all components included. This shows that the inclusion of definitions and candidates offers benefits over classical prompting-based approaches. The results for GPT4 (Table 8) show very similar trends, with again the row corresponding to all components being included having a much better average rank than all other settings. We also see that of all components of the system, the removal of the definitions and the examples are the most damaging across all the datasets.

**Robustness to Example Choice:** To assess the impact of the choice of few-shot examples, we evaluated the GPT models on 5 different randomly selected support sets (3 examples which contain at least one instance of every entity type). The variance of the results suggests that the choice of examples is important (Table 14), however, the average performance across support sets is still high. We also measured the performance when using completely random demonstrations (the 3 examples may not contain instances of several entity types) and noticed a moderate drop in performance of around 5% F1 across datasets.

| GPT4 | Annotator 1 | Annotator 2 | Annotator 3 | Annotator 4 | Annotator 5 |
|---|---|---|---|---|---|
| Conll | 83.4 ± 5.3 | 80.7 ± 7.2 | 78.8 ± 3.6 | 79.1 ± 3.1 | 82.7 ± 6.4 |
| Genia | 58.4 ± 4.5 | 53.6 ± 3.2 | 60.2 ± 4.8 | 54.7 ± 5.6 | 57.1 ± 3.5 |

Table 9: Robustness to different annotators on GPT4

| GPT3.5 | Annotator 1 | Annotator 2 | Annotator 3 | Annotator 4 | Annotator 5 |
|---|---|---|---|---|---|
| Conll | 78.6 ± 4.6 | 72.3 ± 5.1 | 79.1 ± 4.4 | 77.4 ± 3.8 | 73.6 ± 5.6 |
| Genia | 52.8 ± 6.8 | 49.4 ± 7.6 | 50.3 ± 5.2 | 56.2 ± 4.9 | 54.6 ± 5.5 |

Table 10: Robustness to different annotators on GPT3.5.

**Difference in Annotation:** We measure the extent to which the performance of GPT3.5 and GPT4 varies with different annotators. 5 different human annotators were asked to create explanations to complete the FewShot demonstrations for the ConLL dataset. While performance does vary across annotators on different datasets, the average performance on both datasets is still a consistent improvement over previous methods (Table 10). We also try a system where the explanation/ annotations required for the FewShot demonstrations are generated automatically by the LLM. This is a system without any human in the loop, removing any influence of different annotation styles. This system is significantly worse than when human annotation is used (Table 15). Qualitatively, we observe that when using the fully automated system, even PromptNER GPT4 often fails to follow the output format implied in the examples. Frequently, the explanations generated by the LLM spans multiple lines, references multiple entity types and takes non standard formats which leads to the PromptNER output format being highly unstable and hard to reliably parse.

**Prompt Engineering:** We investigate the performance gains achievable when we have access to a validation set that can be used to tune the specific prompts used for each dataset. We used a validation set of 150 data points and manually tuned the definitions used to improve the micro F1-score. We observe improvements on the final test set F1-score across all datasets (6 ). The magnitude of the improvement is dependent on both the dataset and the LLM used. On datasets such as Genia, we achieve improvements as high as 10% (absolute), while on datasets like the AI split of CrossNER we are unable to improve by more than 4%. GPT4 shows stronger performance increases than GPT3.5, owing to its ability to follow more complicated instructions and definitions. Qualitatively, the most successful definition crafting strategies were the ones that identified common trends of mistakes across the validation set and explicitly mentioned a rule in the definition to avoid these mistakes. On ConLL, PromptNER struggled with correctly classifying sports teams named after locations as (ORG) as opposed to (LOC)). Explicitly specifying that if a word that is typically used as a location refers to a sports team, it should be marked as (ORG) significantly reduces these errors.

## 6   HUMAN SURVEY OF ERRORS

To better understand the failure cases of PromptNER, we look to Confusion Matrices (Figure 3). There is a clear trend across all datasets—the not-an-entity tag 'O' is involved in more errors than any other entity type (entity phrase was misclassified as non-entity or non-entity was misclassified as some entity), closely followed by the miscellaneous tags. When inspecting the 'O' errors we found that several times the prediction and ground truth seemed equally valid given the under specified nature of NER problems (see Figure 2 for multiple examples). To quantify the extent to which this is true we conducted a survey where annotators are shown the list of entity phrases identified by PromptNER and ground truth and are required to indicate which list (if any) can be considered better at identifying which words or phrases are named entities. For a complete description of the survey please refer to the Appendix A.1

In all of the domains, at least 20% of the examples are ones where PromptNER has the better (as judged by annotators) entity list. PromptNER is worse than the ground truth label no more than 40% of the time (Table 12). In 4/5 of the domains, the difference between the accuracy of the ground truth labels and PromptNER does not exceed 10%. There is interestingly a domain (Politics)

| Prompt Engineering | ConLL | Genia | Literature | Music | AI |
|---|---|---|---|---|---|
| GPT4 | 88.2 | 66.3 | 80.2 | 88.4 | 67.6 |
| GPT3.5 | 82.4 | 54.7 | 67.8 | 81.3 | 60.8 |

Table 11: Performance when definition in prompt is manually tuned using validation accuracy of 150 points as reference. Manual tuning consistently leads to performance increases.

| Survey | All | Pol | Lit | Music | AI | Sci |
|---|---|---|---|---|---|---|
| PromptNER Acc | 44 | 60 | 30 | 65 | 35 | 30 |
| Labels Acc | 52 | 35 | 80 | 65 | 40 | 40 |
| PromptNER better | 33 | 60 | 20 | 20 | 30 | 35 |
| PromptNER worse | 56 | 15 | 60 | 60 | 65 | 50 |

Table 12: Study of human preference for PromptNER predicted entity list over the ground truth. All numbers are percentages, Kappa Fleis of 0.55. PromptNER Acc and Labels Acc show the fraction of these lists marked as having a correct answer. Better and worse rows show the percentage of times annotators selected PromptNER as the better or worse list. Results show there are many cases where the 'incorrect predictions' of PromptNER should be considered as good as the ground truth.

where PromptNER is considerably better than the ground truth labels. This survey confirms that there are many legitimate errors in the PromptNER predictions, with the ground truth label consistently providing a better option. The survey also shows, however, that even before we consider the entity type annotations, a very considerable percentage of the disagreeing predictions made by PromptNER are considered roughly equally acceptable solutions to the NER problem instance. The results suggest that the F1 metric scores reported in Table 2 underestimate the true performance on the NER task, especially in domains like Politics and Sciences, where the annotators were not likely to say that the PromptNER prediction was worse than the ground truth labels. More generally, the survey exposes a need for human evaluation and a measure of inter-annotator disagreement on NER datasets, as on datasets with high disagreement it may be less fruitful to expect models to approach the 100% F1 mark.

## 7 LIMITATIONS

For a given input, PromptNER predicts phrases that are entities, along with their associated entity types. It does not indicate what the span of a phrase is in the original sentence i.e. the indices of the phrase in the tokenized representation of original sentence. This then necessitates a hand-crafted parsing strategy to infer the spans of the predicted entity phrases from the LLM output. When phrases appear multiple times in the same sentence as different entity types, the parsing strategy can fail to correctly match the various occurrences of the phrase and the various types predicted by PromptNER. There is also a danger of interpreting parts of this pipeline incorrectly for the sake of interpretability—the explanations provided by the system can be logically inconsistent and should not be considered a way to make the NER system interpretable, the candidate lists can completely miss entities from a sentence and so should not be considered as a safe shortlist which will contain all entities with high probability.

## 8 CONCLUSION

We introduce PromptNER, an NER system that outperforms competing few-shot methods on the ConLL, GENIA, FewNERD, FabNER and TweetNER datasets, outperforms cross-domain methods on 3/5 splits of CrossNER, and has superior performance than few-shot and chain-of-thought prompting on GPT4. We conduct a human study of the disagreements between the ground truth of the CrossNER dataset and our model output, finding that a notable percentage of the disagreements are not ones that are considered 'mistakes' by a human annotator. Overall, we provide an alternate way to approach the few-shot NER problem that requires no specialized pretraining, is easy to adjust to different domains, and maintains performance despite using very little training data.

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

# A APPENDIX

## A.1 HUMAN SURVEY DETAILS

The objective of this survey is to quantify the extent to which the list of 'entity phrases' predicted by PromptNER is inferior (with respect to the task of identifying the correct entities) to the ground truth.

### A.1.1 METHODOLOGY

We randomly selected 20 examples from each CrossNER dataset where the prediction and ground truth entity lists differ (the set of entities identified in a sentence, ignoring type) and asked 10 human annotators (each example is seen by 3 distinct annotators) to comment on the lists. Specifically, the annotators are given a dataset dependent definition of the NER problem and are shown two lists of entities. One of these lists is the set of entities identified by our GPT4 PromptNER method, the other is the ground truth, however, evaluators are not given this information. They indicate whether each list has correctly identified all the named entities and only named entities in a sentence, and if not are asked to provide the phrases that are incorrectly identified. Finally, they are asked for an opinion on which list (if any) can be considered better at identifying which words or phrases are named entities. For each dataset there were 5 'toxic' options that trivially have a correct answer (the two lists were completely equal, the trivially correct answer is to not have a preference between the two lists) to serve as attention checks, all annotators passed these attention checks. The exact description of the task is as follows 'Each example will have a definition of what a Named Entity is, a sentence, and two attempts at identifying the correct entities. The task is to answer whether the lists have successfully identified all and only the named entities in the sentence.'

```
Text: Summary of Saturday 's Spanish first division match
Prediction: Spanish first division | True | as it references the
Spanish professional football division, hence it is an organization (ORG)
Ground Truth: Spanish is of type (MISC), the other words are of tag 'O'

Text: The Palestine Liberation Organisation ( PLO ) negotiators said the
last two weeks of talks with Israel on implementing the long-delayed
handover of most of Hebron to PLO rule had been " meaningless " ,
necessitating an Arafat-Netanyahu meeting .
Prediction: Arafat-Netanyahu meeting | True | this is a slightly tricky
case as it refers indirectly to two people, Yasser Arafat
and Benjamin Netanyahu.
Hence, it can be considered as a two entities –
Yasser Arafat (PER) and Benjamin Netanyahu (PER).
Ground Truth: Both words are of tag 'O'

Text: " It shows that Sakakibara is not for a stronger dollar either , "
said Sumitomo 's Note .
Prediction:  's Note | False | it's a possessive phrase; note is not
a person or organization
Ground Truth: Note is tagged PER

Text: The UK Department of Transport on Friday said that the latest
round of " open skies " talks with the U.S. had ended with no deal on
liberalising the transatlantic flight market and no
date set for when talks would restart .
Prediction: The UK Department of Transport | True | It's a named
organization or institution (ORG)
Ground Truth: The is of tag 'O', UK is tagged (LOC) and
Department of Transport is tagged (ORG)
```

Figure 2: Several discrepancies between the prediction and the ground truth are cases where the 'ideal' solution is subjective and not clearly defined by the task definition

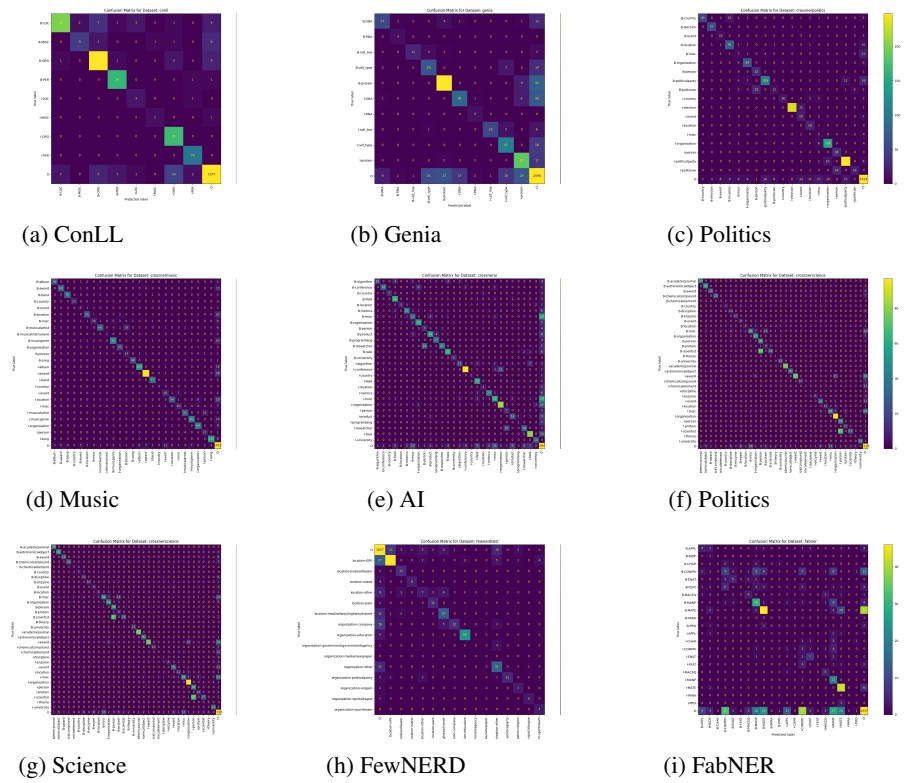

(a) ConLL        (b) Genia        (c) Politics

(d) Music        (e) AI        (f) Politics

(g) Science        (h) FewNERD        (i) FabNER

Figure 3: Confusion Matrices of GPT4 model on sample of 100 datapoints from each dataset. The 'O' class (final column and row) is consistently involved in the most errors. This is seen most clearly on FabNER and Genia

### A.1.2 RECRUITMENT

The annotators for this task were recruited through the personal networks of the author (friends, associates and family). There was no explicit monetary compensation offered or provided for this task. To mitigate the potential affect of personal bias in the annotation they were not informed about the PromptNER tool and its methodology, how the survey would be used in this work or that the two lists in the annotation were from two 'competing' models.

### A.1.3 ETHICAL CONSIDERATIONS

The sentences and entity types do not involve any graphic descriptions, disturbing language pertaining to sensitive topics (including but not limited to hate speech, violence, sexual content etc.) or personal information. However as a safeguard the annotators were informed that should they feel any discomfort or reservations regarding the sentences they encountered they were free to stop the annotation process at any time and report their issues if required. No annotator reported any issues after the task was completed.

| Definitions | Few Shot | CoT | Candidates | ConLL | Genia | Politics | Literature | Music | AI | Science | FewNERD |
|---|---|---|---|---|---|---|---|---|---|---|---|
| ✓ | ✓ | ✓ | ✓ | 83.4 ± 5.6 | **58.4 ± 6.8** | **78.6 ± 4.7** | **74.4 ± 6.2** | **84.2 ± 5.1** | 64.8 ± 6.8 | **72.5 ± 4.1** | **72.6 ± 5.4** |
| ✓ | ✓ | ✗ | ✓ | 78.5 ± 5.2 | 51.8 ± 5.2 | 70.3 ± 3.1 | 69.6 ± 4.1 | 73.8 ± 2.5 | **66.1 ± 4.6** | 67.6 ± 4.1 | 59.5 ± 3.6 |
| ✓ | ✗ | ✓ | ✓ | 67.4 ± 3.4 | 49.2 ± 3.6 | 72.4 ± 5.1 | 63.5 ± 5.6 | 80.5 ± 6.4 | 60.5 ± 4.7 | 59.7 ± 3.5 | 62.8 ± 4.7 |
| ✓ | ✗ | ✓ | ✓ | **84.3 ± 2.4** | 30.2 ± 7.9 | 62.8 ± 5.2 | 53.2 ± 6.7 | 63.5 ± 5.6 | 42.7 ± 3.8 | 30.2 ± 7.9 | 30.2 ± 7.9 |
| ✗ | ✓ | ✓ | ✓ | 70.7 ± 5.8 | 53.3 ± 6.1 | 65.6 ± 4.3 | 55 ± 2.35 | 53 ± 5.8 | 45.1 ± 4.2 | 53.3 ± 6.1 | 43.2 ± 6.3 |
| ✗ | ✓ | ✗ | ✗ | 63 ± 4.72 | 38.5 ± 6.2 | 60.2 ± 6.1 | 54.6 ± 5.7 | 58.9 ± 5.3 | 35.9 ± 4.6 | 46.3 ± 7.5 | 27.1 ± 5.3 |
| ✗ | ✓ | ✗ | ✗ | 66.7 ± 7.0 | 27.4 ± 5.2 | 58.3 ± 5.2 | 46.4 ± 5.1 | 54.7 ± 2.4 | 27.7 ± 6.3 | 36.2 ± 5.3 | 21.9 ± 4.5 |

Table 13: Complete Ablation Results for GPT4

| Random | ConLL | Genia | Pol | Lit | Mus | AI | Sci | FewNERD |
|---|---|---|---|---|---|---|---|---|
| GPT4 | 80.8 ± 7.3 | 62.1 ± 10.2 | 81.6 ± 5.4 | 70.7 ± 8.1 | 86.3 ± 6.7 | 59.2 ± 8.3 | 70.2 ± 4.7 | 75.7 ± 7.6 |
| GPT3.5 | 74.8 ± 11.3 | 54.2 ± 7.4 | 68.7 ± 4.3 | 66.3 ± 7.0 | 73.2 ± 3.8 | 62.5 ± 8.3 | 65.5 ± 8.8 | 59.8 ± 5.1 |

Table 14: Robustness to Randomized selection of FewShot examples, results over 3 different sets of 3 FewShot examples. To obtain these examples we first randomly sample points from the training dataset until there is at least one mention of every entity type and then randomly sample 3 examples from this reduced set

| Automatic | Conll | Genia | Pol | Lit | Music | AI | Sci | FewNerd |
|---|---|---|---|---|---|---|---|---|
| GPT3.5 | 62.72 ± 8.4 | 29.45 ± 7.8 | 32.61 ± 4.2 | 21.45 ± 3.6 | 49.32 ± 10.4 | 14.74 ± 3.4 | 26.82 ± 4.1 | 30.56 ± 5.2 |
| GPT4 | 70.43 ± 10.3 | 41.26 ± 7.7 | 43.52 ± 8.2 | 30.44 ± 4.8 | 68.76 ± 3.8 | 22.53 ± 5.1 | 44.81 ± 7.2 | 48.73 ± 8.2 |

Table 15: Performance of Models with completely automated pipeline (no human annotation required for explanations)

A.2 EXACT PROMPTS USED FOR EACH DATASET

AI

Defn:  An entity is a person(person), country(country), location(location), organisation(organisation), field of Artificial Intelligence(field),
task in artificial intelligence(task), product(product), algorithm(algorithm),
metric in artificial intelligence(metrics), university(university),
researcher(researcher), AI conference (conference), programming language (programlang)
or other entity related to AI research (misc).
Dates, times, adjectives and verbs are not entities.

Example 1: Since the Google acquisition , the company has notched up a number of significant achievements ,
perhaps the most notable being the creation of AlphaGo , a program that defeated world champion Lee Sedol at the complex game of Go

Answer:
1. Google | True | as it is a company or organisation (organisation)
2. creation | False | as it is an action
3. AlphaGo | True | as it is a program or product using AI (product)
4. Lee Sedol | True | as this is a person but not a researcher (person)
5. Go | True | as this is a game that the AI played and is an entitty (misc)

Example 2: In machine learning , support-vector machines ( SVMs , also support-vector networks ) are supervised learning models with learning algorithm s that analyze data used for classification and regression analysis .

Answer:
1. machine learning | True | as it is a field of AI (field)
2. support-vector machines | True | an algorithm in AI (algorithm)
3. SVMs | True | the abbreviation of support-vector machines which is an algorithm (algorithm)
4. supervised learning | True | a subfield of AI (field)
5. learning algorithms | False | as it is not a specific algorithm or task
6. classification | True | as it is a specific task in machine learning or AI (task)
7. regression analysis | True | as it is a specific task in machine learning or AI (task)

Figure 4: AI Input Prompt

Figure 5: Conll Input Prompt

FewNERD

Defn: Entities are locations (location) and organizations (organisation). Entities are not a person (person), piece of art, product, language, living thing, currency, god or scientific concept in astronomy, biology etc. ., the names of buildings and events are also not entities. Dates, times, abstract concepts and adjectives are not entities.

Example 1: Albert Einstein used 100 USD to purchase the Eiffel tower from the Association of Artificial Intelligence

Answer:
1. Albert Einstein | False | as this is the name of a person
2. USD | False | as this is the name of a currency
3. purchase | False | as this is an action or verb
4. Eiffel tower | False | as this is the name of a building
5. Association of Artificial Intelligence | True | as this is an organization (organisation)

Example 2: In England, there is a festival called the Grand Jubilee, founded in 1982 by Attila the Hun, it was the original birthplace of the painting 'The Starry Night'.

Answer:
1. England | True | as it is a location (location)
2. festival | False | as it is not a named entity
3. Grand Jubilee | False | as it is an event
4. 1982 | False | as it is a date
5. Attila the Hun | False | as it is a person
6. The Starry Night | False | as it is a piece of art

Figure 6: FewNERD Input Prompt

GENIA

Defn: An entity is a protein (protein), group of proteins (protein), DNA, RNA, Cell Type (cell_type) or Cell Line (cell_line). " Abstract concepts, processes and adjectives are not entities"

Example 1: In primary T lymphocytes we show that CD28 ligation leads to the rapid intracellular formation of reactive oxygen intermediates ( ROIs ) which are required for CD28 -mediated activation of the NF-kappa B CD28-responsive complex and IL-2 expression

Answer:
1. primary T lymphocytes | True | as they are a kind of cell type (cell_type)
2. CD28 | True | CD28 is one of the proteins expressed on T cells (protein)
3. reactive oxygen intermediates ( ROIs ) | False | as they are not a protein, DNA, RNA, Cell Type or Cell Line
4. NF-kappa B | True | Nuclear factor kappa B (NF-κB) is an ancient protein transcription factor (protein)
5. CD28-responsive complex | True | it is a complex of the protein (protein)
6. IL-2 | True | as it is a protein (protein)

Example 2:
The peri-kappa B site mediates human immunodeficiency virus type 2 enhancer activation in monocytes but not in T cells

Answer:
1. peri-kappa B site | True | as it is a  is a cis-acting element that is a DNA (DNA)
2. human immunodeficiency virus type 2 enhancer | True | as it is a DNA (DNA)
3. Activation | False | as it is a process
4. monocytes | True | as they are a type of cell (cell_type)
5. T cells | True | as they are a type of cell (cell_type)

Figure 7: GENIA Input Prompt

Literature

Defn: An entity is a person(person), country(country), location(location), organisation(organisation), book(book), writer(writer), poem(poem), magazine(magazine), award(award), event(event), country(country), literary genre (literarygenre), nationality(misc) or other enitity in literature (misc).
Dates, times, adjectives and verbs are not entities.

Example 1: The poor conditions of the hospital in Lambaréné were also famously criticized by Nigerian professor and
 novelist Chinua Achebe in his essay on Joseph Conrad ' s novel Heart of Darkness :
 In a comment which has often been quoted Schweitzer says : ' The African is indeed my brother but my junior brother .

Answer:
1. hospital | False | as it is a building type not a named location
2. Lambaréné | True | as it is a location in which the hospital is located (location)
3. Nigerian | True | as it is a nationality (misc)
4. professor | False | as it is not an entity type as defined by the list
5. Chinua Achebe | True | as this is a write who is a novelist (writer)
6. Joseph Conrad | True | as this is a writer who wrote a novel called the Heart of Darkness (writer)
7. novel | True | as this is a genre or type of literature (literarygenre)
8. Heart of Darkness | True | as this is the name of a book (book)
9. Schweitzer | True | as this is a person, not a writer (person)
10. African | True | as this is like a nationality (misc)

Example 2: During this period , he covered Timothy Leary and Richard Alpert ' s Millbrook , New York -based
Castalia Foundation at the instigation of Alan Watts in The Realist , cultivated important friendships with William S. Burroughs and Allen Ginsberg , and lectured at the Free University of New York on ' Anarchist and Synergetic Politics ' in 1965 .

Answer:
1. period | False | as this indicates a time period
2. Timothy Leary | True | as this is a person who has not written a literary work (person)
3. Richard Alpert | True | as this person hasn't written a literary work(person)
4. Millbrook | True | as it is a location inside New York (location)
5. New York | True | as it is a state (location)
6. Castalia Foundation | True | as it is an organisation (organisation)
7. instigation | False | as it is an action
8. Alan Watts | True | as it is a person who has written in a magazine (writer)
9. The Realist | True | the name of a magazine (magazine)
10. William S. Burroughs | True | the name of famous author  (writer)
11. Allen Gibsberg | True | a person who has written literary works (writer)
12. Free University of New York | True | a university is an organisation (organisation)
13. Anarchist and Synergetic Politics | True | some formal academic work (misc)

Figure 8: Literature Input Prompt

Music

Defn: An entity is a person(person), country(country), location(location), organisation(organisation), music genre(musicgenre), song(song), band(band), album(album), artist(musicalartist), musical instrument(musicalinstrument), award(award), event(event) or musical entity (misc)
Dates, times, adjectives and verbs are not entities.

Example 1: Artists from outside California who were associated with early alternative country included singer-songwriters
such as Lucinda Williams , Lyle Lovett and Steve Earle , the Nashville country rock band Jason and the Scorchers and the British post-punk band The Mekons .

Answer:
1. Artists | False | because it is not a specific artist, it is a common noun
2. California | True | it is a state (location)
3. alternative country | True | as it is a musical genre (musicgenre)
4. Lucinda Williams | True | as this is an artist (musicalartist)
5. Lyle Lovett | True | as this is an artist (musicalartist)
6. Steve Earle | True | as this is an artist (musicalartist)
7. Nashville country rock band | True | as it is an entity related to music (misc)
8. Jason and the Scorchers | True | as it is the name of a band not a person (band)
9. British | True | as it is a nationality (misc)
10. post-punk | True | as it is a music genre (musicalgenre)
11. The Mekons | True | as it is a band (band)

Example 2: The film was nominated for the Academy Awards for Academy Award for Best Picture , as well as Academy Award for
Best Production Design ( Carroll Clark and Van Nest Polglase ) , Academy Award for
Best Original Song ( Irving Berlin for Cheek to Cheek ) , and Dance Direction ( Hermes Pan for Piccolino and Top Hat ) .

Answer:
1. Academy Awards | True | as it is an award (award)
2. Academy Award for Best Picture | True | as it is the name of a specific award (award)
3. Academy Award for Best Production Design | True | as it is the name of an award (award)
4. Carroll Clark | True | it is a person but not a musician (person)
5. Van Nest Polglase | True | person but not a musician (person)
6. Academy Award for Best Original Song | True | an award (award)
7. Irving Berlin | True | a person who recieved an award for a song is a musician or artist (musicalartist)
8. Dance Direction | True | an award (award)
9. Hermes Pan | True | a person who is not a musician (person)
10. Piccolino | True | a dance performance name (misc)
11. Top Hat | True | name of a dance (misc)

Figure 9: Music Input Prompt

Science

Defn: An entity is a person(person), university(university), scientist(scientist), organisation(organisation), country(country), location(location), scientific discipline(discipline), enzyme(enzyme), protein(protein), chemical compound(chemicalcompound), chemical element(chemicalelement), event(event), astronomical object(astronomicalobject), academic journal(academicjournal), award(award), or theory(theory). Abstract scientific concepts can be entities if they have a name associated with them. If an entity does not fit the types above it is (misc) Dates, times, adjectives and verbs are not entities.

Example 1: He attended the U.S. Air Force Institute of Technology for a year , earning a bachelor 's degree in aeromechanics , and received his test pilot training at Edwards Air Force Base in California before his assignment as a test pilot at Wright-Patterson Air Force Base in Ohio .

Answer:
1. U.S. Air Force Institute of Technology | True | as he attended this institute is likely a university (university)
2. bachelor 's degree | False | as it is not a university, award or any other entity type
3. aeromechanics | True | as it is a scientific discipline (discipline)
4. Edwards Air Force Base | True | as an Air Force Base is an organised unit (organisation)
5. California | True | as in this case California refers to the state of California itself (location)
6. Wright-Patterson Air Force Base | True | as an Air Force Base is an organisation (organisation)
7. Ohio | True | as it is a state (location)

Example 2: In addition , there would probably have been simple hydride s such as those now found in gas giants like Jupiter and Saturn , notably water vapor , methane , and ammonia .

Answer:
1. hydride | True | as it is a chemical (chemicalcompound)
2. gas giants | True | as it is a category of astronomical object (misc)
3. Jupiter | True | as it is a planet (astronomicalobject)
4. water vapor | True | as it is a chemical (chemicalcompound)
5. methane | True | as it is a chemical (chemicalcompound)
6. ammonia | True | as it is a chemical (chemicalcompound)

Figure 10: Science Input Prompt

Politics

Defn: An entity is a person(person), organisation(organisation), politician(politician), political party (politicalparty), event(event), election(election), country(country), location(location) or other political entity (misc). Dates, times, abstract concepts, adjectives and verbs are not entities

Example 1: Sitting as a Liberal Party of Canada Member of Parliament ( MP ) for Niagara Falls , she joined the Canadian Cabinet after the Liberals defeated the Progressive Conservative Party of Canada government of John Diefenbaker in the 1963 Canadian federal election .

Answer:
1. Liberal Party of Canada | True | as it is a political party (politicalparty)
2. Parliament | True | as it is an organisation (organisation)
3. Niagara Falls | True | as it is a location (misc)
4. Canadian Cabinet | True | as it is a political entity (misc)
5. Liberals | True | as it is a political group but not the party name (misc)
6. Progressive Conservative Party of Canada | True | as it is a political party (politicalparty)
7. government | False | as it is not actually an entity in this sentence
8. John Diefenbaker | True | as it is a politician (politician)
9. 1963 Canadian federal election | True | as it is an election (election)

Example 2: The MRE took part to the consolidation of The Olive Tree as a joint electoral list both for the 2004
European Parliament election and the 2006 Italian general election , along with the Democrats of the Left and
Democracy is Freedom - The Daisy .

Answer:
1. MRE | True | as it is a political party (politicalparty)
2. consolidation | False | as it is an action
3. The Olive Tree | True | as it is a group or organization (organisation)
4. 2004 European Parliament election | True | as it is an election (election)
5. 2006 Italian general election | True | as it is an election (election)
6. Democrats of the Left | True | as it is a political party (politicalparty)
7. Democracy is Freedom - The Daisy | True | as it is a political party (politicalparty)

Figure 11: Politics Input Prompt

