# OpenReview forum: "PromptNER : Prompting For FewShot Named Entity Recognition"
_ICLR.cc/2024/Conference — Submitted to ICLR 2024_

### Official Review · Reviewer_CmJa · 2023-10-28

**Soundness:** 2 fair
**Presentation:** 2 fair
**Contribution:** 1 poor
**Rating:** 3
**Confidence:** 5

**Summary:**

This paper introduces PromptNER, a prompt-based algorithm for few-shot and cross-domain Named Entity Recognition (NER). It utilizes a Large Language Model (LLM) to generate potential entities along with explanations for their compatibility with entity types. The method requires modular entity type definitions, few-shot examples, and explanatory text. PromptNER achieves better performance on few-shot NER and cross-domain NER over baseline methods. The study demonstrates the flexibility and adaptability of PromptNER across domains with minimal computational cost.

**Strengths:**

This work emphasizes the importance of prompt-based heuristics and in-context learning in advancing few-shot learning for NLP. It presents a promising approach for flexible and easy-to-apply NER systems that can adapt to different settings with limited human involvement.

**Weaknesses:**

1. Concerning the introduced PromptNER methodology, I didn't discern any noteworthy technical novelty. In essence, the idea that the paper alludes to strikes me as rather naive. Almost every LLM-based NER initiative could effortlessly incorporate the strategy of embedding entity definitions within prompt texts, facilitating the LLM's comprehension. In a nutshell, I'm unconvinced that this paper heralds any significant fresh technological insights.

2. The related work section falls short of explicitly highlighting how the proposed method differentiates from, and potentially improves upon, existing works. Such delineations are crucial for situating a new method in the context of established research.

3. There's a palpable omission of numerous standard few-shot NER benchmarks, such as WNUT-2017, MIT-Movie and MIT-Restaurant, JNLPBA, among others. Moreover, there's a conspicuous absence of comparative analysis with many of the state-of-the-art few-shot NER models.

4. Of paramount concern is the author's decision to juxtapose their model, based on GPT3.5/GPT4, directly with the baselines. This presents an overtly skewed comparison, heavily favoring the presented model. Such disparities render any conclusions derived from the experiments to be questionably validated. It's evident that OpenAI's LLM models outclass the baselines by multiple magnitudes, and the proprietary nature of these models further impedes reproducibility. A more equitable comparison would necessitate evaluating the proposed prompt methodology and baseline techniques using identical open-source LLMs, ensuring both type and size compatibility.

**Questions:**

1. Given the stated weaknesses, how do the authors justify the claimed novelty of the PromptNER methodology, especially in light of its evident simplicity and seemingly naive approach?

2. Could the authors amplify on the differentiators between their proposed method and existing literature, providing a clearer positioning of their research contribution?

3. What rationale underpins the selection of benchmarks for evaluation, and why were certain prominent few-shot NER benchmarks omitted?

4. Can the authors elucidate their choice to compare their model based on GPT3.5/GPT4 directly with the baselines, given the glaring discrepancies in terms of model scale and accessibility?

---

> ### Author Response · Authors · 2023-11-21
>
> We thank the reviewer for sharing their comments and concerns. We believe that the simplicity of the PromptNER approach is something that enables it to perform well in diverse scenarios without the need for much modification. To the best of our knowledge, we are the first to present a system that uses a prompting template to perform end-to-end Named Entity Recognition in a FewShot setting without also requiring some form of prefix tuning or parameter updates. It is certainly true that the improvements of PromptNER come from using more powerful LLMs like GPT4. The scaling experiments also confirm that this improvement does not occur when using smaller models. However, in exchange for requiring a much larger model, this method now requires significantly less NER data that matches the target distribution. As for the rationale for the selecting of evaluation benchmarks and methods, we made an effort to test a wide range of 6 different NER benchmarks (ConLL, Genia, FewNERD, CrossNER, FabNER, TweetNER), in an effort to get standard FewShot NER benchmarks (ConLL), NER benchmarks in specialized domains (Genia, CrossNER, FabNER, TweetNER) as well as NER benchmarks which have contradictory information between train and test domains (FewNERD), we compare against 3-7 other methods from the past 2 years and make sure we include the method that reports the highest performance on each dataset. For the FabNER and TweetNER tasks, we selected the best methods from the ConLL dataset results to manually compute the results.

---

### Official Review · Reviewer_2EDv · 2023-10-29

**Soundness:** 3 good
**Presentation:** 3 good
**Contribution:** 1 poor
**Rating:** 5
**Confidence:** 4

**Summary:**

The paper introduces a prompting template for the NER tasks. With GPT3.5 and GPT4, it can achieves the SOTA on a list of few-shot NER tasks measured by F1 scores. The paper also achieves the SOTA performance on 3/5 Cross Domain NER tasks. The prompting techniques uses few-short in-context learning and chain-of-thoughts. In the ablation study, the paper shows that the size of the LLM and few-shot examples are critical to the performance gains.

**Strengths:**

The paper introduces a very simple prompting template which can be easily integrated into relevant applications.
The paper is clearly written and is easy to follow. it gives the detailed ablation study to help us understand the contribution of each components.
The paper compares against a comprehensive list of previous work in its experiments.

**Weaknesses:**

It seems to me that the major contribution of the SOTA performance comes from GPT4 more than some advanced prompting techniques of the paper where most of cases, the best performance is achieved only by GPT 4. T5 is way worse than the current SOTA. The prompting template is simply an application of CoT with few shot in-context learning. I am not convinced if whether this prompt template is very novel or has a significant originality of ideas and I was wondering whether other similar template can't achieve similar performance.

**Questions:**

For the Cross Domain NER tasks, as Table 2 shows, only 2 examples are used in prompting and the F1 scores for AI and Sciences are not the highest. Why can't we add more examples (e.g. up to 200) to improve performance? Did we also consider fine-tune GPT to see if we can get the higher F1 scores?

---

> ### Author Response · Authors · 2023-11-21
>
> We thank the reviewer for their comments. To address their questions, we were limited by the context window length and so could place only a few examples in the prompt. We did not explore the idea of fine-tuning GPT, this is because we wanted to keep the method a FewShot one that uses fewer than 5 examples.

---

### Official Review · Reviewer_Zo5E · 2023-11-01

**Soundness:** 2 fair
**Presentation:** 2 fair
**Contribution:** 2 fair
**Rating:** 3
**Confidence:** 4

**Summary:**

This paper proposed a PROMPTNER framework integrating entity definitions and few-show learning in a large language model prompt. Experiments on GPT-4 LLM show the proposed model is better on several datasets.

**Strengths:**

This paper demonstrated that with the GPT4 as the backbone, the proposed PROMPTNER showed good cross-domain NER identification ability. Several ablation experiments showed the effectiveness of each component.

**Weaknesses:**

There are various studies on improving prompt strategies in this LLM area.  Adding the entity definition on the prompt of LLM is not an innovative method. Based on the experiment, the most improvement of the proposed method comes from the powerful GPT 4 backend. Comparing the GPT4-driven model with models with much weaker LLM is not necessary and not fair (Table 2-4).

**Questions:**

None

---

> ### Author Response · Authors · 2023-11-21
>
> We thank the reviewer for their comments and for sharing their concerns with the comparisons made in the paper. It is certainly true that the improvements of PromptNER come from using more powerful LLMs like GPT4. The scaling experiments also confirm that this improvement does not occur when using smaller models. However, in exchange for requiring a much larger model, this method now requires significantly less NER data that matches the target distribution. This can be seen in our CrossDomain NER experiments, where PromptNER is able to outperform other methods while using less than 2% of the data than other methods. When comparing with other FewShot methods, most other methods utilize a source domain NER dataset that is similar to the FewShot target domain (e.g. pretraining on OntoNotes before transferring onto CONLL). PromptNER does not require the backbone language model to have seen any NER data and hence does not rely on the target domain being similar to the source domain to achieve good performance.

---

### Official Review · Reviewer_9We2 · 2023-11-03

**Soundness:** 3 good
**Presentation:** 3 good
**Contribution:** 2 fair
**Rating:** 3
**Confidence:** 3

**Summary:**

The paper proposed to use LLM to perform named entity recognition which is widely studied in the academia of natural language processing. The proposed method involves the definition of the entity types, few-shot demonstrations, and chain-of-thought templates for generation. The experiments results show a great improvement compared  with baselines on NER tasks and cross-domain NER tasks.

**Strengths:**

Since LLMs are quite popular these days, it is worth to see how LLMs can be used in the classic NLP tasks. This work explores the potential of LLMs in NER tasks and show they are useful in terms of cross-domain and low-resource scenarios.

**Weaknesses:**

The biggest concern is that the method seems straightforward to me, and thus I think it lacks the core innovation in terms of the methodology. It is intuitive to inform the model of the definition, few-shots, and chain-of-thought to accomplish the task. I wish to see how these prompts are interacting with the final outputs so as to provide more insights on how the future work can learn from the prompt design or use the LLMs properly in classic NLP tasks. Also, I think it is more innovative to design specialized modules in the prompt engineering for NER tasks. Otherwise, the current work is similar to a strong baseline which is helpful to future work for sure, but may not be ready for a long research paper.

**Questions:**

See weaknesses.

---

> ### Author Response · Authors · 2023-11-21
>
> We thank the reviewer for their comments. We do agree that the methodology and its design is intuitive; we believe that it is a strength of the method that would enable its use in a wide range of NER problem domains without requiring much change. While developing the method, we did explore the option of having a system that operated using specialized modules instead (e.g. a system that looped over every word/phrase to classify them independently and a system that went through multiple rounds of refinement using the prompt), however, the performance of the system presented in the paper was superior.

---

### Meta-Review · Area_Chair_DMAc · 2023-11-26

**Metareview:**

This paper proposed PromptNER framework, which integrates entity definitions in the design of prompting template to achieve cross-domain Named Entity Recognition (NER) with few shot in-context learning and chain-of-thoughts. PromptNER utilizes a Large Language Model (LLM) to generate potential entities along with explanations for their compatibility with entity types. With GPT3.5 and GPT4, PromptNER achieves SOTA on a list of few-shot NER tasks measured by F1 scores, as well as 3 out of 5 Cross Domain NER tasks. In the ablation study, the paper shows that the size of the LLM and few-shot examples are critical to the performance gains - improvement can be seen when using powerful LLMs like GPT4 but does not occur when using smaller models.

Strength:
 - The paper works on popular topic of LLMs and show LLMs (GPT4) can be useful in the classic NLP tasks (NER) at cross-domain and low-resource scenarios. Ablation experiments show the effectiveness of each component and the entire methodology.
 - The paper introduces a very simple prompting template which can be easily applyed to NER systems.

Weakness:
 - The paper focus on prompt engineering and explores the better design of prompt for NER. There are already many works in this direction, and prompt engineering has already been proved effective in various tasks and NLP/multimodal/speech areas. There are lack of novel findings in this work except confirm again the powerful of GPT4 in NER tasks. This paper might fit well in a language focused conference. However, as ICLR community is looking for generic scientific breakthroughs, this paper seems of limited interests to the community.
 - PromptNER yields performance gain over SOTA only when larger models (GPT4) are applied but not for smaller models (e.g., T5) indicates that most benefit comes from GPT4, and this paper only contributes a way to better leverage GPT4 at NER task. This contribution is relatively limited for a paper in ICLR venue.

**Justification For Why Not Higher Score:**

As mentioned in the weakness of this paper, this paper only works on prompt engineering for NER tasks. The topic is of limited interests to ICLR community and contributions are limited. Thus I recommend to reject this work.

**Justification For Why Not Lower Score:**

N/A

---

### Decision · Program_Chairs · 2024-01-16

Reject